# Review: How Forage Feeding Early in Life Influences the Growth Rate, Ruminal Environment, and the Establishment of Feeding Behavior in Pre-Weaned Calves

**DOI:** 10.3390/ani10020188

**Published:** 2020-01-22

**Authors:** Jianxin Xiao, Gibson Maswayi Alugongo, Jinghui Li, Yajing Wang, Shengli Li, Zhijun Cao

**Affiliations:** 1State Key Laboratory of Animal Nutrition, Beijing Engineering Technology Research Center of Raw Milk Quality and Safety Control, College of Animal Science and Technology, China Agricultural University, Beijing 100193, China; dairyxiao@gmail.com (J.X.); maswayi@yahoo.com (G.M.A.); yajingwang_cau@163.com (Y.W.); lisheng0677@163.com (S.L.); 2Department of Animal Science, University of California, Davis, CA 95616, USA; lgreyhui@hotmail.com; 3State Key Laboratory of Animal Nutrition, College of Animal Science and Technology, China Agricultural University, Beijing 100193, China

**Keywords:** calves, forage, performance, rumen fermentation, behavior

## Abstract

**Simple Summary:**

Under natural grazing systems, calves are likely to consume forage in early life. However, forage inclusion in the diet of pre-weaned calves has long been a controversial issue due to it possibly being associated with negative calf performance. Recent published literature seems to confound previous research. This review aims to understand the factors that may influence forage inclusion in the ration of pre-weaned calves. We have explored research related to the effect of feeding forage on rumen and behavioral development to better understand whether forage should be fed to the young calf. Based on the findings, it is concluded that a small amount of good quality forage is recommended for calves to improve their behavioral expression and rumen environment, which may further improve calf performance.

**Abstract:**

The provision of forage to pre-weaned calves has been continuously researched and discussed by scientists, though results associated with calf growth and performance have remained inconsistent. Multiple factors, including forage type, intake level, physical form, and feeding method of both solid and liquid feed, can influence the outcomes of forage inclusion on calf performance. In the current review, we summarized published literature in order to get a comprehensive understanding of how early forage inclusion in diets affects calf growth performance, rumen fermentation, microbiota composition, and the development of feeding behavior. A small amount of good quality forage, such as alfalfa hay, supplemented in the diet, is likely to improve calf feed intake and growth rate. Provision of forage early in life may result in greater chewing (eating and ruminating) activity. Moreover, forage supplementation decreases non-nutritive oral and feed sorting behaviors, which can help to maintain rumen fluid pH and increase the number of cellulolytic bacteria in the rumen. This review argues that forage provision early in life has the potential to affect the rumen environment and the development of feeding behavior in dairy calves. Continued research is required to further understand the long-term effects of forage supplementation in pre-weaned calves, because animal-related factors, such as feed selection and sorting, early in life may persist until later in adult life.

## 1. Introduction

As early as 1897, researchers began to evaluate hay feeding in young calves [1]. Since then, more calf related studies involving various aspects such as genetics, nutrition, health, and welfare have been completed [2]. Likewise, over the last hundred years, the use of forage in pre-weaned calves has remained one of the most key concerns in calf nutrition.

Before the 1950s, forage feeding was generally encouraged in pre-weaned calves, as it was believed to reduce abnormal behavior (e.g., eating bedding material) [3], lower diarrhea [4], and improve rumen development [5,6]. However, new research emerged challenging the fact that forage feeding could improve rumen development to the same degree as calf starter [7]. Volatile fatty acids (VFA) were considered to play a more critical role in stimulating rumen epithelial development rather than the physical form of diets [8]. Specifically, forage ration resulted in a higher proportion of acetate [9], which did not stimulate the growth of rumen papillae to the same extent as butyrate and propionate [10,11]. Concentrates, high in rapidly fermentable carbohydrates, produced more butyrate and propionate [12]. Therefore, higher proportions of concentrates could enhance the development of rumen papillae [13]. Furthermore, as fiber had lower digestibility than starch and sugar, many studies claimed that roughage increased gut fill because of low ruminal fermentation rate, thus curbing the consumption of starter feed which had higher energy density [5,14,15]. Therefore, some dairy farms provided calves with ad libitum access to concentrate feed, with no forage until after weaning [16]. More recently, in the 2000s and 2010s, more studies have investigated the effects of forage feeding in pre-weaned calves, yet results were inconsistent. Some of the studies reported a decrease [15,17], an increase [18,19,20,21,22,23], or no differences [24,25,26,27] in solid dry matter intake (DMI) and average daily gain (ADG) when forage was added in the calf diets. As the solid DMI and growth rate of the pre-weaned calves are important factors that drive rumen development [15] and subsequent milk production in the first lactation [28], it is vital to understand the factors that influence feed consumption and calf growth when forage is added to their diet.

On the other hand, over the last ten years, more research has explored the effects of forage inclusion in diets of young calves not only on calf performance and rumen morphological development but also on ruminal fermentation metabolites, bacterial composition, and feeding behavior [2], giving us a more comprehensive and better understanding of this topic. Therefore, the aims of this review are: (1) to discuss the factors that contribute to the inconsistent results in performance in calves with forage inclusion; (2) to summarize and evaluate the latest literature on the role of forage in rumen fermentation and the establishment of feeding behavior of calves.

## 2. Factors that Affect Calf Performance with Forage Inclusion

Generally, under natural grazing systems, adult dairy cows spend 7 to 13 h eating grass every day [29]. Young calves acquire nutrients from both milk and fresh grasses and begin to graze as early as week 2 of age [30,31]. The grazing time usually lasts for a short period, around 20 min at 10 days, which increases rapidly to 360 min at 100 days of age, equivalent to 70 percent of the grazing time in adult cattle [30,32].

On some commercial dairy farms, calves are offered free access to starter feed before weaning without forage [33], which is contrary to natural grazing. Forage inclusion in the pre-weaned calf diet has long been discouraged due to its negative effect on the growth rate [15,17]. However, recent research has shown that several factors need to be taken into account when evaluating the impact of forage provision on calf performance [8,9,34]. These factors include the source, amount, particle size, physical form, offering time, and feeding method of forage and concentrate, as well as the amount of milk offered and milk feeding method. A summary of these studies is presented in Table 1, Table 2, Table 3, Table 4, Table 5 and Table 6.

### 2.1. Forage Factors

Forage provision in pre-weaned calf remains a controversial topic, hence the proliferation of related research. In earlier studies, it was believed that forage was the main component in calf diets that played an essential role in rumen development [5,6]. In two different studies offering a high proportion of roughage (80% vs. 60% vs. 40% and 67% vs. 50%) to calves as hay to grain respectively, Hibbs et al. [5] and Conard and Hibbs [35] found that DMI and ADG increased as the proportion of concentrate in the ration increased. Stobo et al. [36] obtained similar results with calves provided a maximum daily allowance of concentrate at 0.45, 0.91, 1.36, 1.81, or 2.27 kg/d along with ad libitum access to grass hay (crude protein (CP) = 7.5%, crude fiber (CF) = 28.7%). These authors reported that as the concentrate intake increased, the hay intake decreased linearly [5,36], while live and empty body weight (BW) and rumen development were improved [36]. This is probably because concentrates result in more VFA production, especially propionate and butyrate [12], which enhances papillae development [13,36]. Collectively, these early studies suggested that in a high roughage feeding system, the addition of a bulky forage in the diet might decrease the consumption of energy-dense concentrates, leading to less rumen fermentation and lower degradation rates, and subsequently lower total nutrient intake and calf growth.

In the early 2000s, several studies began to investigate the effect of lower levels of forage inclusion in the diet on pre-weaned calf performance. Most of them included a proportion of forage ranging between 5 and 25% of total solid feed intake [15,19,22,27,37,38,39,40]. In contrast to previous studies, most studies either reported an increase [19,22,38,40] or a lack of differences [27,37] in DMI, ADG, and BW, indicative of multifactorial effects in these studies.

#### 2.1.1. Forage Level and Source

Coverdale et al. [19] conducted two experiments in which starter supplemented with relatively low level (7.5 and 15%) of bromegrass hay appeared to improve DMI, ADG, and feed efficiency (FE). In experiment 1, limited amounts of mixed feed (concentrate and forage) were offered before weaning, followed by ad libitum feeding post-weaning. Calves receiving coarse starter with either 7.5 and 15% of bromegrass hay (8 to 19 mm) were heavier and had greater ADG and FE than calves receiving only coarse starter, while calves fed 7.5% of hay tended to have the highest ADG and FE [19]. In experiment 2, calves were offered diets ad libitum and weaned according to intake. The concentrate and total DMI tended to be higher in calves fed 7.5 and 15% of bromegrass hay when compared with the non-forage group [19]. Similarly, Hosseini et al. [40] recently reported that compared to non or 15% straw, feeding 7.5% of chopped wheat straw tended to improve the overall total solid feed intake (659, 685, and 826 g/d, respectively) and ADG (519, 553, and 620 g/d, respectively) when calves were offered 4 L of whole milk per day. Feeding alfalfa hay at 10% of total solid feed increased the overall DMI, ADG, and final BW, and thereby shortened the time to weaning at a target DMI of starter (1 kg for 3 consecutive days), compared with feeding 0 or 5% of alfalfa hay [22]. Nemati et al. [38] also observed a linear increase in total DMI and ADG during the postweaning (d 52 to 70) and overall periods (d 3 to 70) of calves supplemented with chopped alfalfa hay at 0, 12.5, and 25% on dry matter (DM) basis. However, gut fill could be a confounding factor when evaluating the effect of forage feeding on improving ADG in dairy calves. It is commonly believed that an increase in ADG and BW in calves fed forage could be due to greater gut fill [15,20,27]. Therefore, there is a need for further investigation of the relationship between gut fill and ADG.

Poor performance has also been commonly observed when including forage in the diet [15,17]. Hill et al. [15] reported that feeding either 2.5 and 5% of chopped timothy hay linearly reduced starter intake, ADG, empty body weight ADG, and FE. The quality of forage significantly influences the digestibility and the palatability of the diet [46]. Ülger et al. [47] compared two calf total mixed ration (TMR) diets with either 20% of a high-quality alfalfa hay (CP = 18.1%, acid detergent fiber (ADF) = 36.1% and neutral detergent fiber (NDF) = 44.4% on DM basis, relative feed value (RFV) = 127.2) or lower quality wheat straw (CP = 3.7%, ADF = 52.4% and NDF = 80.1% on DM basis, RFV = 55.9) and found that the high-quality roughage improved FE and numerically increased ADG during the preweaning period. In a more recent study, Hill et al. [48] found that moderate to low-quality grass hay (5.4% CP and 62.8% NDF on DM basis) reduced the digestibility of DM, OM, and CP in young calves consuming a textured starter. It is noteworthy that the type of hay was not specified in this study, and we speculated that timothy or mixed hay was included based on other studies at the same period by the authors [49]. On the contrary, Castells et al. [21] and Hosseini et al. [40] reported that a low-quality straw (CP = 4.2%, and NDF = 74.0% on DM basis) could also improve DMI and ADG. The inconsistency in results on calf performance when providing low-quality hay was likely due to the different amount of milk offered, which may affect the solid feed consumption and forage preference as discussed below.

Castells et al. [21] evaluated ad libitum provision of different types of forages in the diets from 2 weeks of age and found that the inclusion of alfalfa hay (CP = 16.6%, ADF = 30.2%, and NDF = 40.2% on DM basis), rye-grass hay (CP = 6.8%, ADF = 35.1%, and NDF = 59.3% on DM basis), oat hay (CP = 8.4%, ADF = 31.8%, and NDF = 59.6% on DM basis), barley straw (CP = 4.2%, ADF = 42.5%, and NDF = 74.0% on DM basis), corn silage (CP = 8.6%, ADF = 25.2%, and NDF = 41.9% on DM basis), or triticale silage (CP = 7.5%, ADF = 42.3%, and NDF = 64.7% on DM basis) resulted in similar or increased intake and gains without impairing FE and nutrient digestibility. Increased DMI and ADG were observed when oat hay, barley straw, or triticale silage were offered, and the inclusion of alfalfa hay did not exhibit similar benefits, probably due to the preference for this high-quality and palatable forage [21]. Indeed, forage intake was highest in calves fed alfalfa hay (14% of total solid DM) compared with the other treatments (oat hay: 8% rye-grass hay: 4%, barley straw: 5%, corn silage: 5%, and triticale silage: 4%). The higher proportion of forage to concentrate ratio may limit the DM digestibility and hence restrict the DMI and ADG, as we have already discussed [5,35]. More recently, a meta-analysis by Imani et al. [34] evaluating the effect of forage provision on growth performance of dairy calves using 27 published studies from 1998 to 2016 revealed that concentrate DMI was higher in calves offered alfalfa hay compared with those offered other types of forages.

#### 2.1.2. Forage Physical Forms and Processing

The focus is not only on the forage source and level of feeding, but also on the physical form and processing of forage. Mirzaei et al. [27] evaluated the effects of particle size (short at 2.92 mm vs. long at 5.04 mm as geometrical means) of alfalfa hay on growth performance of dairy calves at two different inclusion rates (low at 8% vs. high at 16% on DM basis). The authors observed no differences in growth rates between calves fed with or without hay, but greater DMI and weaning BW were found in calves fed low levels of alfalfa with a long particle size (8% and 5.04 mm) and high levels with a short particle size (16% and 2.92 mm) compared with calves fed low levels with a short particle size (8% and 2.92 mm) and high levels with a long particle size (16% and 5.04 mm). The short particle size at a low level of alfalfa might not have the potential to increase the capacity, motility, and development of the rumen [11,60], while the negative effect on performance by the long particle size at a high level might be attributed to the lower digestibility rate of long particles compared with short particles [61]. Montoro et al. [62] also found that when calves were supplemented with 10% of long chopped (3 to 4 cm) ryegrass hay, DMI, ADG, and FE were greater than those fed 10% finely ground (2 mm) grass hay. Longer particle size improved performance [8], probably because of the increased rumination time of calves, which increased saliva production, and consequently improved buffering effect on the ruminal environment [63,64]. However, inconsistent results have been obtained by Omidi-Mirzaei [65] and Suárez-Mena [66]. Omidi-Mirzaei et al. [65] reported that when calves were fed forage with different particle size (alfalfa hay: short = 1.96 mm or long = 3.93 mm; and wheat straw: short = 2.03 mm or long = 4.10 mm as geometric mean), rumination time increased in calves fed forage with long particle size, but concentrate DMI, ADG, and FE were not affected. Suárez-Mena et al. [66] compared four different particle sizes (0.82, 3.04, 7.10, and 12.7 mm as geometric mean) of low-quality forage (5% straw) mixed in the diet and observed no effect on DMI, growth performance, and minimal changes in rumen fermentation and pH among treatments. In summary, these results imply that interactions may exist among forage source, level, and particle size, and the optimal inclusion level of forage should be determined based on the forage source and particle size.

In recent years, attempts have been made on alternative ways to increase solid feed consumption of dairy calves, such as using non-forage fiber [39], silage based feed [55,67], moisturized starter [68,69], and reconstituted hay [58,59,70]. Beet pulp is a common source included in the diet as a non-effective fiber source. Maktabi et al. [39] observed that 10% of beet pulp in the diet improved DMI and ADG compared to a control group (no fiber inclusion), but growth was not enhanced when 20% of beet pulp was used. Inclusion of corn silage early in life of dairy calves has recently gained more interest [55,67]. In an experiment that compared supplementing 15 against 0% of corn silage, DMI, ADG, and BW increased probably because of the higher moisture content of corn silage that contributed to reduced dustiness and increased palatability of the feed [55,67]. However, feeding a high level of corn silage (30 or 60%; 75 or 100%) offered no benefits compared with feeding concentrate alone [43,44]. Suárez et al. [43] reported that substitution of 30 or 60% of the concentrate by corn silage did not affect DMI and ADG but feeding 30% of straw reduced DMI. Kehoe et al. [44] also found no differences in DMI and ADG when including 0, 75, or 100% of corn silage in pre-weaned calf diet but fed solely corn silage diet stunted the growth of rumen papillae and tended to impair intestinal morphology. Hence, it is possible that corn silage can be used to partly replace the concentrate with little harmful effects on the growth and development of the calf.

It has been documented that moisturizing the concentrate starter feed by adding water to change the DM from 90 to 50% increased DMI, ADG, and VFA production in dairy calves [68,69]. More recently, hay processing by reconstituting with water was evaluated in a series of studies published by Kargar et al. [58,59,70,71]. Hay was soaked in water for 24 h and mixed every 6 h to obtain a theoretical DM content of 20% [71]. This method has been used previously in the diets of mature cows to increase fecal consistency [72] and reduce digestion lag time in the rumen as a result of a higher fiber digestibility [73]. Kargar et al. [70,71] replaced dry alfalfa hay (10%) with a similar amount of reconstituted alfalfa hay, resulting in similar DMI and ADG, but higher NDF digestibility during the pre-weaned period. Furthermore, a greater improvement in health status (fecal score and general appearance score) was obtained with reconstituted hay, possibly due to decreased dustiness, similar to corn silage [70]. Therefore, corn silage, reconstituted hay, and beet pulp can be used interchangeably in dairy calf diets based on availability and the relative feed price [58,59].

#### 2.1.3. Time and Method of Offering Forage

Time [24,75,76] and method [25,77,78] of offering forage are essential factors that can influence how dairy calves utilize forage. While Wu et al. [24] found no differences in DMI, ADG, and rumen development in calves fed alfalfa hay or oat hay either at day 3 or 15 of age, different outcomes were shown in two other studies, investigating the effect of age at which alfalfa hay [75] and oat hay [76] were introduced to calves. Both studies observed improved DMI and growth performance with forage provision, and the greatest growth performance and rumen development were obtained in calves offered hay from the 2nd week rather than the 4th or 6th week of age [75,76]. Based on these studies, suggestions may be made to include alfalfa or oat hay in diets of calves as early as week 2 or even right after birth in order to improve DMI and ADG in dairy calves [24,75].

Forage have been provided as a mixture with concentrate in previous studies, while recent studies investigated calf preference to different feeds by providing forage and concentrate separately. Castell et al. [21] observed a greater DMI and ADG in calves provided forage rather than without forage. In this study, calves consumed around 5% of forage when it was offered ad libitum and separately from the concentrate. However, several studies did not observe a positive effect on DMI when part of the concentrate was substituted for forage before weaning [43,47]. In research by Ülger et al. [47], who mixed forage at 20% with the concentrate and Suárez et al. [43] at either 30 or 60%, calves consumed at a predetermined fixed forage to concentrate ratio which was much greater compared to calves fed free choice. Therefore, the greater forage proportion in the mixture diet (i.e., containing 30 or 60% forage) [43] might mask the positive effect on DMI and growth performance compared with forage consumed voluntarily (around 5% forage) by calves [21]. A meta-analysis study has proven this inference; when forage was offered separately, starter feed intake and ADG increased compared to a mixed ration diet [34]. On the contrary, some studies claimed that DMI and growth performance were not different between two feeding methods (mixed vs. separate) [77,78], possibly due to the low proportion of forage (15 or 10%) in mixed ration, which was similar to that (11 or 10%) with separate forage provision in studies by Overvest [77] and EbnAli [78]. Moreover, the high level of milk feeding (around 13 or 26% of birth weight) might have decreased the solid feed consumption [77,78]. Although the forage feeding method may not always lead to better performance, it certainly affects the expression and development of dairy calf behavior [25,79], as discussed below.

### 2.2. Concentrate and Milk Factors

#### 2.2.1. The Physical Form of Concentrate Feed

There are different forms and types of calf starters. Porter et al. [37] reported that whether forage was included or not, calves on coarse mash (average particle size: 2014 μm) ate and gained more than those on pelleted diets (average particle size: 888 μm). Moreover, rumination was initiated earlier. Hence, up to 8 weeks of age, calves raised on a complete concentrate diet without forage did not experience a significant depression in growth performance, which might be due to the long particle size of coarse mash feed initiating rumination early and preventing bloat and parakeratosis in the rumen [65]. Two consecutive experiments were conducted by Terré et al. in 2005 to evaluate the influence of the physical form of concentrate feed (textured or pelleted) with or without forage inclusion on the performance of young calves. Calves receiving pelleted concentrate feed with straw exhibited a greater solid intake and higher rumen fluid pH compared with those receiving a pelleted concentrate feed without straw. Calves that received the texturized (containing whole corn) starter feed had equivalent rumen fluid pH to those fed a pelleted concentrate with straw. However, rumen fluid pH and performance were not improved when another texturized (containing rolled mixed grains) concentrate feed was offered [80]. These results show that the physical forms of concentrate feed may affect the calf performance and rumen environment differently. For example, calves fed a texturized concentrate feed containing whole corn had a greater rumen fluid pH than steam-flaked corn, dried-rolled corn, and roasted-rolled corn [81], likely because the calves spent a longer time chewing the whole corn feed, which increased saliva production, hence neutralizing the rumen pH and acids. In another study involving a mashed (with or without corn silage) and a textured concentrate (with or without corn silage), regardless of the physical form of concentrate feed, forage inclusion resulted in greater DMI, ADG, and final BW than non-included calves [67]. It was evident that forage provision had more effect on the growth performance than the physical form of the feed, whereby steam-flaked grains were the main component in the concentrate. In agreement, Mojahedi et al. [82] reported that including alfalfa hay could improve DMI and ADG of calves fed steam-flaked corn, as opposed to a cracked corn-based diet, probably because of higher amounts of gelatinized starch in the steam-flaked corn (44.1 vs. 12.5% of total starch, respectively). Possibly, forage inclusion enhanced starch fermentability of the steam-flaked corn through the provision of effective fiber. Collectively, a decrease in solid feed consumption in calves fed finely ground or pelleted starter on commercial farms compared with those fed textured concentrate [83] warrants forage provision to improve the solid feed intake, growth performance, and rumen environment to a greater extent [80,82].

#### 2.2.2. The Amount and Method of Milk Feeding

Most studies that suggested exclusive concentrate feeding were conducted with calves fed low amounts of milk [15,36]. For example, in the Hill et al. [15] study, only 120 L of milk was fed to calves before 28 days of life (weaning date), averaging around 4 L/d of milk (approximately 10% of birth body weight) which is insufficient for optimal growth. This low milk feeding rate might stimulate greater concentrate intake in calves to make up for the deficit in energy requirements. Indeed, a strong negative correlation between liquid and concentrate feed intake has been elucidated in a meta-analysis that shows calves fed high milk or milk replacer resulted in limited daily starter intakes [28].

As discussed earlier, compared to concentrates, forage are bulkier and are less digestible and have lower fermentation rates [5], which can lead to a low voluntary intake when low energy forage is offered separately or as a mixture with the concentrates [8,86]. Castells et al. [21] claimed that when calves were offered different forages (hay, straw, and silages, respectively) with concentrates ad libitum and separately, forage consumption was only 4–6% of the total solid feed intake. Interestingly, the proportion of hay consumed across studies seems to range from 3 to 45% of total solid feed intake [20,21,25,52,87]. The difference in the proportion of forage consumed across multiple studies may also depend upon milk feeding amounts. In two studies feeding different amounts of milk, Castells et al. [52] reported that calves consumed 3% of total solid feed as forage at a low level of milk feeding (214 L from d 0 to d 57, averaged 4 L/d, around 10% of birth body weight), while Xiao et al. [52] reported a greater ratio of forage to total solid feed intake, approximately 45% when a high amount of milk was offered (376 L from d 0 to d 56, averaged 6.8 L/d, around 17% of birth body weight) [25]. Milk contains a high content of fat and sugar, which provide the energy required by the calves, and greater milk amounts might alter concentrate requirements [25]. This speculation concurs with another study in which calves fed low amounts of milk consumed more concentrates, resulting in a lower ratio of forage to total solid feed intake in a low compared to a high milk feeding group (13.2% vs. 18.6%) [84].

Feeding patterns and methods could also affect forage intake in calves. When investigating the effect of either step down (fed at around 15% of birth body weight per day) or conventional (approximately 10% of birth body weight per day) feeding patterns in dairy calves, Khan et al. [88] found that the former had better performance. In a different study, Daneshvar et al. [23] reported that when similar amounts of milk were fed using different feeding patterns (step down vs. conventional), solid feed intake did not differ between treatments. Horvath et al. [85] showed that the feeding method (bucket vs. teat feeding) did not alter the forage and concentrate intake. Hence, milk allowance might have a greater impact on solid feed consumed by calves as opposed to the milk feeding pattern or method.

Limited studies directly investigating the relationship between milk allowance and forage consumption in pre-weaned calves are available, which calls for scientists to turn their attention to this area, especially with more farms leaning towards high milk volume feeding. Forage inclusion can promote total solid feed consumption and BW gain in calves, but factors such as the amount of forage, forage sources, forage feeding method, physical form of forage and concentrate, and milk allowance might confound these benefits. Calves should be slowly introduced to relatively low levels of forage while guarding against the use of low digestible forage (i.e., straw), which may depress total DMI and BW gain. Moreover, forage should be available free-choice and in separate containers from concentrate feed.

Table 7 shows a summary of selected studies that determined the effects of forage inclusion on performance, rumen fermentation and development, and expressive behavior in dairy calves. While the effects of feeding forage on performance, such as DMI and ADG, were controversial, relatively consistent results were obtained in other parameters, like rumen fermentation and expressive behavior.

## 3. Rumen Environment

### 3.1. Rumen Fluid pH and Fermentation

The rumen is the largest and most crucial compartment of the digestive system in adult ruminants, as it is vital for acquiring metabolic substrates through microbial fermentation. Although young calves have an undeveloped rumen, fermentation begins at a very early age [89] and may affect the development and health of the rumen. Prolonged low rumen fluid pH may cause subacute ruminal acidosis (SARA) in adult cows, which is well defined in beef feedlot cattle (pH < 5.8 for 3 h/d) [90] and dairy cows (as periods of moderately depressed pH, from about 5.0 to 5.5) [91]. However, in young calves, ruminal acidosis has not been clearly defined. Previous studies reported that rumen fluid pH in young calves is often well below 5.8 [89,92]. Some researchers believe that, as in mature cows, dairy calves can experience ruminal acidosis [34,89], probably due to the high amount of concentrate fed [92] in artificially rearing systems and the relatively low saliva [93] secreted at a young age.

Concentrate feed, high in rapidly fermentable carbohydrates, such as sugar and starch, provides energy for optimal growth, but the fermentation rate tends to generate lots of VFA and lactic acid, resulting in low rumen fluid pH [51]. Forage, high in fiber, may play a role in mitigating this challenge. Most of the studies (21 studies, accounting for 84% of summarized studies) explored in this review showed a positive effect of forage inclusion on rumen fluid pH in dairy calves, while very few reported no difference (four studies, accounting for 16% of summarized studies) or negative impact (Table 7). In agreement with a previous meta-analysis, our literature search showed that forage could improve the rumen fluid pH when supplemented to calves, though it might be dependent on the forage source [38,40]. Alfalfa hay is more likely to modulate rumen fluid pH during the milk-feeding period than other types of forages [33]. Maktabi et al. [39] reported that increasing fiber content by adding beet pulp (10% and 20%) in the concentrate diet failed to improve rumen fluid pH, while supplementing alfalfa hay (10%) resulted in a significant improvement in this parameter by providing more effective NDF. Terré et al. [53] also demonstrated that increasing NDF content (18.2 vs. 26.7%) by adding soybean hulls in the pelleted starter could not alter rumen fluid pH, but adding chopped oat hay, containing more effective fiber, could improve the ruminating behavior resulting in a higher pH. In agreement, Laarman et al. [94] reported a positive relationship between forage intake and rumen fluid pH, while SARA (rumen fluid pH below 5.8) could be exacerbated when calves are fed less than 0.08 kg/d [64], suggesting that even small amounts of forage consumption (timothy hay, 0.08 kg/d) can reduce rumen acidosis in calves.

The possible reasons for the increased rumen fluid pH when adding forage in the diet are multifactorial. On the one hand, forage is bulkier and has lower digestibility compared to the concentrate. The higher forage consumption leads to increased intake of effective fiber, which in turn stimulates the chewing (ruminating and eating) activity of calves [56,95], and subsequently improves the saliva production and rumen buffering [38,63]. On the other hand, the rapidly fermentable carbohydrates generate abundant VFA that may exceed and overwhelm the absorptive capacity of the undeveloped rumen [96]. Feeding forage in pre-weaned calves could reduce the concentration of VFA [19,23,25,35,40,52,53,54,67] and decrease rumen plaque formation [42,52], increasing the absorptive surface area of the rumen epithelium and hence reducing the accumulation of VFA and maintaining the appropriate rumen fluid pH. In addition, an increased passage rate in the rumen was observed in calves fed forage compared to those fed concentrate only, which lowered the feed retention time in the gastrointestinal tract (28.4 h for concentrate feed vs. 18.8 h for oat hay group), fermentation time and VFA concentration [52]. In the same study, calves fed forage tended to have a higher expression of monocarboxylate transporter-1 in the rumen wall [52], which plays a central role in transporting acetate, lactate, and protons from the rumen lumen to the bloodstream [97,98], hence alleviating VFA accumulation in the rumen as well.

A greater acetate [22,23,40,43,52,53,67], and a lower propionate [25,34,40,43,54,67], butyrate [23,25,27,39,40,53], and valerate [23,25,52,53,67] concentration/proportions have been reported in calves fed with forage than those fed only concentrate. These dynamics in fermentation patterns are probably related to the changes in the rumen microbial ecosystem. For example, cellulolytic microbes, such as *Ruminococcus flavefaciens* and *Ruminococcus albus*, are more prevalent in animals fed high forage diets, which increase fiber degradation and elevates the proportion of acetate in the rumen [42,52]. Both propionate and butyrate stimulate and enhance rumen epithelial development [12,13,99], with butyrate serving as the preferred energy source as well as modulating the gene expression in the rumen epithelium [96]. A low proportion of these two VFA may limit the growth of rumen papillae [42,52]. Due to its relatively low proportions, the valerate has received little attention. It has been suggested that cellulolytic microbes utilize valerate in the rumen [100], which might explain its decrease in calves supplemented with forage in their diets.

Lactic acid decreases in the rumen when forage is included in the diet [51], which might end up positively altering rumen fluid pH. Terré et al. [53] found an interesting relationship between rumen VFA and rumen fluid pH. When rumen fluid pH was above 5.1, total VFA and rumen fluid pH were linearly correlated; however, when it fell below 5.1, the correlation disappeared. The implications are that lactic acid (a much stronger acid than VFA) may alter rumen fluid pH at a pH below 5.1 [53,101]. In adult cows, an acute ruminal acidosis was observed, with excessive consumption of concentrate feed leading to a sudden and uncompensated drop in rumen fluid pH (below about 5.0). Owens et al. [60] showed that lactic acid concentrations increased with a decline in rumen fluid pH. However, when rumen fluid pH (around 5.0–5.5) was moderately depressed, lactic acid accumulation was inconsistent [102] or transiently fluctuated [103]. Hence, although moderate depression in rumen fluid pH may cause SARA in dairy cows, it is not because of the lactic acid accumulation, but might be due to the accumulation of VFA alone [102].

Collectively, previous studies demonstrate that forage provision has a positive effect on rumen fluid pH and alters rumen fermentation in calves. However, the majority of these studies assessed rumen fluid pH and VFA at only a single time point. Further research is encouraged to test the dynamic changes in rumen fluid pH and VFA when different types, amounts, timing, and particle sizes of forage are supplemented in the diet, which may help us define SARA in young calves more accurately.

### 3.2. Rumen Microbes

A developed rumen is full of microorganisms that ferment and degrade multiple nutritional fractions (sugar, starch, fiber, protein, fat, and so on) and provides necessary metabolic substrates and nutrients to the dairy cattle. The microbial ecosystem differs between young calves and adult cows [104]. At birth, young calves possess no anaerobic microorganisms in the rumen [105], with recent evidence suggesting that colonization occurs immediately after birth [104,106]. Dominant microbes that are involved in normal rumen function of mature cows are present as early as one day of age [106]. This colonization of microorganisms and the presence of substrates trigger fermentation activity, which then provides indispensable nutrients for rumen development. It might take as long as a year for the rumen to mature and for calves to establish a stable rumen microbiota system with many factors involved [104].

Both liquid and solid feed appear to affect the microbial community in young calves [107]. In this review, our discussion is restricted to the effect of forage feeding on the microbiota of young calves. Castells et al. [52] reported that forage supplementation (alfalfa) numerically increased cellulolytic microbes (*Ruminococcus albus*) compared with calves fed only concentrate feeds. Similarly, Kim et al. [54] observed significantly higher copy numbers of cellulolytic bacteria (*Ruminococcus flavefaciens* and *Ruminococcus albus*) in calves supplemented with forages (orchard and timothy hay). Early studies also evaluated the effects of the physical form (finely ground, 1 mm theoretical particle size vs. unground, 0.64 cm theoretical particle size) of diet on rumen microbiota with two identical diets (25% alfalfa hay and 75% grain) that varied only in particle size. Calves offered the ground diet had a relatively lower rumen fluid pH and lower number of cellulolytic bacteria than calves fed the unground diet [89]. These results revealed that effective fiber might play a crucial role in changing the rumen environment, hence altering the microbial populations.

The next-generation sequencing (NGS) analysis has revealed that the major phyla in the rumen are *Firmicutes* (around 43%), *Bacteroidetes* (around 21%), *Actinobacteria* (around 18%), and *Proteobacteria* (around 4%) [108]. Relatively higher abundance of *Bacteroidetes* and lower abundance of *Actinobacteria* was observed in calves supplemented with forage compared to those fed only concentrate [54]. *Bacteroidetes*, the second most dominant phyla in calves’ rumen, may stimulate the development of the digestive tract [109]. In adult cows fed concentrate feed, the relative abundance of *Bacteroidetes* dropped and cows were more susceptible to SARA [110]. The lower level of *Bacteroidetes* in calves fed only concentrate could be partly explained by greater feed intakes resulting in low rumen pH. Furthermore, Kim et al. [54] found that the most dominant genus in *Bacteroidetes* phylum was *Prevotella*, a highly active hemicellulolytic and starch-degraders [111] that mainly produce acetate. A relatively higher abundance of *Prevotella* may be related to the greater acetate proportion in calves offered forage [54]. Jami et al. [104] reported that *Prevotella* was the predominant genus in animals fed high-fiber diets rather than high-caloric diets. *Olsenella* is an important lactic acid-producing bacterium under the phylum *Actinobacteria* [112]. Forage inclusion decreases *Olsenella* (hay, 3.9% vs. concentrate, 13.2%) significantly, which contributes to the lower abundance of *Actinobacteria* (hay, 4.7% vs. concentrate, 13.9%) [54]. Thus, it can be speculated that forage inclusion in the diet might affect the growth of lactic acid-producing bacteria (such as *Olsenella*) by limiting the proportion of rapidly fermentable substrates (e.g., starch) replaced by fiber. In contrast, although numerical differences were observed in *Bacteroidetes* and *Actinobacteria* when evaluating the effect of forage supplementation, Lin et al. [108] reported that neither alpha nor beta diversity indices and microbiota were significantly different among the dietary groups [108], possibly because of the volume of milk (Lin, 252 L, 10% vs. Kim, 88 L, 4% of birth bodyweight) fed to calves, resulting in varying solid feed consumed. Alternatively, different forage sources (Lin, Oat hay vs. Kim, Timothy) and feeding levels (Forage/Total Solid: Lin, 6% vs. Kim, 20%) (Table 2 and Table 4) might have led to insufficient forage consumption causing the changes in the composition of rumen microbiota in the Lin et al. study. These results indicate that the population of most predominant microbiota (e.g., *Bacteroidetes* and *Actinobacteria*) in the rumen is closely related to the amount and type of solid feed consumed. Further studies need to focus on these major groups to illustrate the relationship between rumen microbiota, fermentation and feed consumption in dairy calves.

It is important to note that the effect of various nutritional (sugars, starch, rumen degradable protein, NDF, or ADF) and physical (effective forage fiber or non-forage fiber) fractions on rumen microorganisms in young calves is not clearly defined in literature. Furthermore, most of the studies evaluating microbial and molecular changes in young calves have only studied a small number of microbes of interest. With the rapid development of the NGS technologies in the past 20 years, we recommend the exploration of the global changes in microbial abundance to better understand substrate fermentation and absorption and epithelium development in the rumen as forage are supplemented to calves in early life.

## 4. Feeding Behavior

### 4.1. Ruminating and Eating Behavior

Feed experiences and behavior development in early life might affect the behavioral expression of adult ruminants [79]. In the last decade, researchers have increased their attention on the development of calf behaviors, such as eating, standing, lying, and ruminating when forage is included in the diet. Forage inclusion in the diet undoubtedly increases chewing in calves even before weaning when only a small amount of solid feed is consumed. Increased chewing activity may be as a result of higher rumination [21,39,40,53,55,56,75,76,77,78] or the total time spent eating [39,40,57,76,77] when calves are fed forage.

In newborn ruminants, rumination is initially absent and emerge a few weeks after birth [113]. Providing forage to young calves can accelerate the development of rumination behavior [114,115]. van Ackeren et al. [116] observed that chewing time was lower in calves receiving a low NDF diet (26.2%) compared with those receiving a high NDF diet (31.3%). Porter et al. [37] claimed that calves began ruminating by week 4 of age when fed a more physically effective solid feed, while those fed a finely pelleted feed began ruminating from week 6. Rumination is crucial in ruminants helping maintain the rumen fluid pH by stimulating saliva production that neutralize VFA and lactic acid in the rumen and thus to maintain a healthy rumen environment [105].

Meal feeding patterns (meal size, frequency, and duration) can also impact the rumen environment. Generally, rumen fluid pH declines rapidly after feed ingestion, and the rate of decrease is associated with meal size and feeding frequency [117]. Large meal sizes and infrequent meals may result in a greater drop in rumen fluid pH post-ingestion. Horvath et al. [57] illustrated that the provision of forage not only increased the total eating time but also influenced the solid feed meal patterns. An improved meal frequency and duration were observed in their study, which leads to relatively slower post-prandial drops in rumen fluid pH, potentially decreasing the risk of SARA [63].

### 4.2. Sorting Behavior

Feed sorting is well demonstrated in adult cows, since they are highly sensitive to sweet taste [118]. Probably, the preference for sweetness reflects the inclination towards higher energy demands, hence the tendency to sort out for concentrates (sweet, high energy-density) in a total mixed ration [119]. The sorting out of the mixed ration can result in an unbalanced nutrient intake, whereby cows sort out for the rapidly fermentable cereals as opposed to forage, leading to a drop in rumen fluid pH, and hence inducing SARA [120]. Ingesting excessive fermentable concentrate feed can result in rumen acidosis [63]. In turn, the sorting behavior is altered further, leading the animals to choose the part of diets with longer particle size and slower fermentable rate [121]. These results suggest that ruminants develop feed preferences based on post-ingestive feedback [122] and they may be biased towards choosing certain nutrients as the situation demands.

Feed sorting is also seen in the early life of calves. When feeding concentrates and forage free-choice, variation in the proportion of forage to total solid intake was observed (ranging from 5 and 45%) [21,25,87]. The changes in dietary selection across multiple studies may depend on forage related factors and milk feeding allowance, as has been discussed above. It is interesting to note that feed preference and sorting can be established early and persist later on in life. Miller-Cushon et al. [123] reported that calves fed either concentrate or forage before weaning were likely to consume the feed that they were already familiar with, even when switched to a mixed diet after weaning. Similarly, our research group found that calves are likely to eat feed they were originally introduced to and familiar with even after switching to a free-choice diet, though this effect only lasted for a short period. However, after switching the diet at weaning, the provision of both concentrate and hay separately early in life led to a greater hay intake ratio (35.6%) than providing concentrate (17.7%) or hay (16.5%) solely before weaning. Furthermore, exposure to a diet of both concentrate and hay early in life could numerically improve the calves’ ability to sort for long particles 6 months later [25]. Therefore, these results suggest that early exposure to feed experience can affect the feed preference immediately after switching diets and may have a long-lasting effect. The feeding method may also play an essential role in influencing the learning of feed sorting behavior. When we compared three different feeding methods (solely concentrate, separated concentrate and forage, mixed concentrate and forage for the first month; data unpublished) in 2 month old calves, the lowest sorting activity was observed in calves fed concentrate and forage separately. Hence, calves exposed early to a diet of concentrate or mixed ration are likely to sort for fine-grain particles, probably because these calves have already established their sorting behavior, which can last even after changing to a new mixed diet (data unpublished). Similarly, the provision of solid feed in pre-weaned calves as separate components reduced the extent of feed sorting after weaning compared to offering the diet as a mixed ration [87]. As already stated, feed sorting is likely to influence rumen fluid pH and may lead to SARA. Separately feeding different solid feed components at the same time may avert sorting for fine particles when compared to feeding solely concentrate or mixed diets and might lead to a more stable rumen fluid pH and a healthier rumen in the calf. However, we cannot ignore the fact that the effect of forage inclusion on sorting behavior is dependent on a myriad of other factors (e.g., forage source, level, physical form, dry matter, and milk allowance).

### 4.3. Other Behaviors

Access to forage by dairy calves may also reduce the occurrence of other non-nutritive oral and abnormal behaviors [21,40,53,55,56,57,76,85], such as tongue rolling, licking of buckets, pen or surface, sniffing, vocalizing, and eating the bedding material. Horvath et al. [85] demonstrated that providing forage decreased the non-nutritive oral behaviors, and when combined with feeding milk by teat, the effects were more significant compared with bucket feeding. These results further buttress the fact that liquid and solid feeding can influence the development of pre-weaned calf behaviors. Furthermore, supplementation of good quality forage increased other behaviors that may indicate satisfaction (tail swishing, self-grooming, and rubbing) [56]. Worthy to note is that the decline in non-nutritive oral behavior may have also reduced the formation of hair and fiber balls in the rumen [4], which have been associated with poor health and growth of calves. Further research is encouraged to explore whether forage inclusion early in life would have a long-term effect on sorting and other behaviors.

## 5. Conclusions

Understanding factors that influence responses to forage inclusion in pre-weaned calves is of significant importance from a management point of view because the effect of offering forage on calf feed intake and growth rate has been inconsistent. In recent studies, a small amount of good quality forage such as alfalfa supplemented in the diet is likely to improve the DMI and ADG. However, these performances are dependent on the type of concentrate and the amount of milk offered. Although controversy remains on whether forage improves growth rate, it has been well documented that its inclusion early in life can help with the establishment of feeding behavior, leading to greater rumination and eating behavior as well as lowering the abnormal feed sorting behavior. All these positive effects can result in a higher rumen fluid pH and a more stable rumen environment, with a corresponding positive effect on rumen microbiota and fermentation. Further research is required to understand the long-term effects of offering forage to pre-weaned calves, since animal-related factors, such as feed selection and sorting, established early in life may persist later on in life.

## Figures and Tables

**Table 1 animals-10-00188-t001:** A summary of studies feeding different levels of forage in pre-weaned dairy calves.

Objectives	Trt ^1^	Calf/Trt	Weaning Age (d)	Forage Feeding Age (d)	Forage Source	Cutting Length/Processing ^2^	Solid Feed Offering Method	Concentrate Physical Form	Amount of Milk Fed ^3^	Outcomes ^4^		Reference
Forage (%)										DMI	ADG	
0, 40, 60, 80%	3	7	49	4	Alfalfa and Timothy hay	-	TMR	-	-	**N ****	**N ****	*Hibbs* et al., 1956 [5]
50, 67%	4	10	49	3	Grass legume silage	-	TMR	Coarse	-	-	-	*Conard* et al., 1956 [35]
5 to 60%	10	4	-	56	Barley, Rye, Wheat straw	-	-	-	-	-	-	*Jahn* et al., 1970 [41]
20–70%	2	6	-	7	Alfalfa hay	-	TMR	—	-	-	-	*Žitnan* et al., 1998 [42]
0, 7.5, 15%	4	16	31	-	Bromegrass hay	Chopped 8 to 19 mm	TMR	Coarse, Ground	-	***p* ***	***p* ***	*Coverdale* et al., 2004 [19]
0, 30, 60%	8	8	70	10	Corn silage, Straw, Grass, Corn silage	-	TMR	Pellet starter	-	**N ****	**NS**	*Suárez* et al., 2007 [43]
0, 16%	4	16	28	3	Beet pulp	-	TMR	Pellet	80 L (Around 7%)	**NS**	**NS**	*Porter* et al., 2007 [37]
0, 5%	2	24	31–32	3–4	Cottonseed hull	GMPL: around 2 mm	TMR	Texture	100 L (Around 10%)	***p* ***	**NS**	*Hill* et al., 2008 *(Trail 1)* [15]
0, 5, 10%	4	12	28	3–4	Cottonseed hull, Timothy hay	GMPL: around 2.2 mm	TMR	Texture	120 L (Around 10%)	**N ****	**N ****	*Hill* et al., 2008 *(Trail 2)* [15]
0, 2.5, 5%	3	16	28	3–4	Timothy hay	GMPL: around 2.2 mm	Free Choice	Texture	120 L (Around 10%)	**N ****	**N ****	*Hill* et al., 2008 *(Trail 3)* [15]
0, 5, 10%	6	7	53	3	Alfalfa hay	GMPL: 2.6 mm	TMR	Finely ground	Around 10%	***p* ****	***p* ****	*Beiranvand* et al., 2014 [22]
8, 16%	5	10	51	16	Alfalfa hay	GMPL: 2.92 vs. 5.04 mm	TMR	Ground	190 L (Around 10%)	-	-	*Mirzaei* et al., 2015 [27]
0, 12.5, 25%	4	15	51	3	Alfalfa hay	GMPL: 3 mm	TMR	Finely ground	204 L (Around 10%)	***p* ****	***p* ****	*Nemati* et al., 2016 [38]
0, 75%, 100%	3	15	56	1	Corn silage	-	TMR	Texture	416 L (Around 18%)	**NS**	**NS**	*Kehoe* et al., 2019 [44]

^1^ Trt = Treatment. ^2^ Geometric mean particle length (GMPL) refers to geometric mean particle length, which was determined using American Society of Agricultural and Biological Engineers (ASABE) forage sieve methods (ANSI/ASAE S424.1) [45]. ^3^ Amount of milk fed is the total amount fed during the whole trial period, while milk feeding rate equals the average daily milk allowance/birth body weight (%). ^4^ Outcomes = effect of forage inclusion, ** indicates a significant effect (*p* < 0.05), * indicates a tendency (*p* < 0.1), N indicates a negative effect, *p* indicates positive effect, and NS shows no significant effect. DMI and ADG were evaluated by comparing calves fed with or without forage.

**Table 2 animals-10-00188-t002:** A summary of studies feeding different forage sources in pre-weaned dairy calves.

Objectives	Trt ^1^	Calf/Trt	Weaning Age (d)	Forage Feeding Age (d)	Forage (%)	Cutting Length/Processing ^2^	Solid Feed Offering Method	Concentrate Physical Form	Amount of Milk Fed ^3^	Outcomes ^4^		Reference
Forage Source										DMI	ADG	
Alfalfa hay, Cottonseed	3	24	-	7	25% of Cottonseed, Ad libitum of alfalfa hay	Chopped to 10 cm	Free choice	-	Around 7%	**NS**	**NS**	*Anderson* et al., 1982 [50]
Beet pulp, Soybean hulls, Corn grits	5	32	84	-	0, 30.3, 46.4, 91.3%	-	TMR	Pellet	608 L (around 18%)	***p* ****	***p* ****	*Suárez* et al., 2006 [51]
Straw; Corn silage, Dried grass	8	8	70	10	-	Chopped	TMR	Pellet starter;	-	**N ****	**NS**	*Suárez* et al., 2007 [43]
Beet pulp	4	16	28	3	0, 16%	-	TMR	Pellet	80 L (Around 7%)	**NS**	**NS**	*Porter* et al., 2007 [37]
Cottonseed hull	2	24	31–32	3–4	0, 5%	GMPL: around 2 mm	TMR	Texture	100 L (Around 10%)	***p* ***	**NS**	*Hill* et al., 2008 *(Trail 1)* [15]
Cottonseed hull, Timothy hay	4	12	28	3–4	0, 5, 10%	GMPL: around 2.2 mm	TMR	Texture	120 L (Around 10%)	**N ****	**N ****	*Hill* et al., 2008 *(Trail 2)* [15]
Timothy hay	3	16	28	3–4	0, 2.5, 5%	GMPL: around 2.2 mm	Free Choice	Texture	120 L (Around 10%)	**N ****	**N ****	*Hill* et al., 2008 *(Trail 3)* [15]
Alfalfa hay	6	7	53	3	0, 5, 10%	GMPL: 2.6 mm	TMR	Finely ground	Around 10%	***p* ****	***p* ****	*Beiranvand* et al., 2014 [22]
Alfalfa hay	4	15	51	3	0, 12.5, 25%	GMPL: 3 mm	TMR	Finely ground	204 L (Around 10%)	***p* ****	***p* ****	*Nemati* et al., 2016 [38]
Alfalfa hay, Ryegrass hay	3	20	57	14.1 ± 4.2	Ad libitum	Chopped	Free choice	Pellet	214 L (Around 9.6%)	**NS**	**NS**	*Castells* et al., 2012 *(Trail 1)* [21]
Oat hay, Barley straw,	3	20	57	14.1 ± 4.2	Ad libitum	Chopped	Free choice	Pellet	214 L (Around 9.6%)	***p* ****	***p* ****	*Castells* et al., 2012 *(Trail 2)* [21]
Triticale silage, Corn silage	3	20	57	14.1 ± 4.2	Ad libitum	Chopped	Free choice	Pellet	214 L (Around 9.6%)	***p* ****	***p* ****	*Castells* et al., 2012 *(Trail 3)* [21]
Alfalfa hay; Oat hay	3	5	56	3	Ad libitum	Chopped	Free choice	Pellet	214 L (Around 10%)	**NS**	**NS**	*Castells* et al., 2013 [52]
Oat Hay	4	16	51	9 ± 4.4	Ad libitum (4, 5%)	Chopped	Free choice	Pellet	152 L (Around 10%)	***p* ****	***p* ****	*Terré* et al., 2013 [53]
Orchard hay, Timothy hay	2	8	56	42	0, 20%	-	-	-	88 L (Around 4%)	**NS**	-	*Kim* et al., 2016 [54]
Wheat straw, Alfalfa hay	2	15	56	14	20%	Chopped to 1–2 cm	TMR	Pellet	212 L (Around 10%)	-	-	*Ülger* et al., 2017 [47]
Alfalfa hay, Beet pulp	2	13	50	4	0, 10, 20%	-	TMR	Texture	228 L (Around 11%)	***p* ***	***p* ****	*Maktabi* et al., 2016 [39]
Alfalfa hay, Corn silage	6	10	49	3	0, 15%	GMPL: 2.9 and 12.07 mm	TMR	Fine ground	196 L (Around 10%)	***p* ****	***p* ****	*Mirzaei* et al., 2017 [55]
Fresh Ryegrass	4	6	49	7–10	Ad libitum	Chopped to approximately 4 cm	Free choice	Pellet	-	**NS**	**NS**	*Phillips* et al., 2004 [56]
Coastal Bermuda grass hay	2	16	56	17 ± 3	Ad libitum (15%)	Chopped to 5 cm	Free choice	Pellet	426 L (Around 19%)	**NS**	**NS**	*Horvath* et al., 2019 [57]
Grass hay	4	4	42	3	Ad libitum	Long (without details)	Free choice	Texture	182 L (Around 10%)	**NS**	**NS**	*Hill* et al., 2019a [48]
Corn silage, Reconstituted alfalfa, Reconstituted beet pulp	3	18	49	3	10%	GMPL: alfalfa, 5 mm and corn silage, 12–15 mm	TMR	Ground	283 L (Around 14.5%)	-	-	*Kagar* et al., 2019 [58,59]

^1^ Trt = Treatment. ^2^ GMPL refers to geometric mean particle length, which was determined using ASABE forage sieve methods (ANSI/ASAE S424.1) [45]. ^3^ Amount of milk fed is the total amount fed during the whole trial period, while milk feeding rate equals the average daily milk allowance/birth body weight (%). ^4^ Outcomes = effect of forage inclusion, ** indicates a significant effect (*p* < 0.05), * indicates a tendency (*p* < 0.1), N indicates a negative effect, *p* indicates positive effect, and NS shows no significant effect. DMI and ADG were evaluated by comparing calves fed with or without forage.

**Table 3 animals-10-00188-t003:** A summary of studies related to forage particle size and methods of processing forage in pre-weaned dairy calves.

Objectives	Trt ^2^	Calf/Trt	Weaning Age (d)	Forage Feeding Age (d)	Forage Source	Forage (%)	Solid Feed Offering Method	Concentrate Physical Form	Amount of Milk Fed ^3^	Outcomes ^4^		Reference
Cutting Length/Processing ^1^										DMI	ADG	
Chopped (GMPL, 5.4 mm) vs. Pelleted (GMPL, 5.8 mm)	3	11	76	3	Alfalfa hay	0, 10%	TMR	Semi-texture	500 L (17%)	**NS**	**NS**	*Jahani-Moghadam* et al., 2015 [26]
GMPL: 2.92 vs. 5.04 mm	5	10	51	16	Alfalfa hay	8, 16%	TMR	Ground	190 L (Around 10%)	-	-	*Mirzaei* et al., 2015 [27]
GMPL: 0.82, 3.04, 7.0, 12.7 mm	4	10	56	1	Straw	5%	TMR	Pellet	Around 12%	-	-	*Suárez -Mena* et al., 2015 [66]
GMPL: Alfalfa (1.96 and 3.93 mm) vs. Wheat straw (2.03 and 4.10 mm)	4	12	49	1	Alfalfa; Wheat straw	Ad libitum	Free choice	Texture	279 L (Around 14%)	-	-	*Omidi-Mirzaei* et al., 2018 [65]
Chopped to 20 to 40 cm	2	24	42	2–3	Timothy hay (*Phleum pratense*)	Ad libitum	Free choice	Texture	178 L (Around 10%)	**NS**	**NS**	*Hill* et al., 2019b [49]
Chopped 3 to 4 cm vs. ground to 2 mm	2	10	49	5	Ryegrass hay	10%	TMR	Crumb	200 L (Around 9%)	-	-	*Montoro* et al., 2013 [62]
Non-forage fiber	4	16	28	3	Beet pulp	0, 16%	TMR	Pellet, Mash	80 L (Around 7%)	**NS**	**NS**	*Porter* et al., 2007 [37]
Non-forage fiber	2	13	50	4	Alfalfa hay, Beet pulp	0, 10, 20%	TMR	Texture	228 L (Around 11%)	***p* ***	***p* ****	*Maktabi* et al., 2016 [39]
Silage based feed	6	10	49	3	Alfalfa hay, Corn silage;	0, 15%	TMR	Fine ground	196 L (Around 10%)	***p* ****	***p* ****	*Mirzaei* et al., 2017 [55]
Silage based feed	4	12	56	3	Corn silage	0, 15%	TMR	Mash, Texture	291 L (Around 13%)	***p* ****	***p* ****	*Mirzaei* et al., 2016 [67]
Silage based feed	3	15	56	1	Corn silage	0, 75%, 100%	TMR	Texture	416 L (Around 18%)	**NS**	**NS**	*Dill-McFarland* et al., 2019 [74]
Silage based feed	3	15	56	1	Corn silage	0, 75%, 100%	TMR	Texture	416 L (Around 18%)	**NS**	**NS**	*Kehoe* et al., 2019 [44]
Reconstituted hay	3	18	49	3	Corn silage, Reconstituted alfalfa, Reconstituted beet pulp	10%	TMR	Ground	283 L (Around 14.5%)	-	-	*Kagar* et al., 2019 [58,59]

^1^ GMPL refers to geometric mean particle length, which was determined using ASABE forage sieve methods (ANSI/ASAE S424.1) [45]. ^2^ Trt = Treatment. ^3^ Amount of milk fed is the total amount fed during the whole trial period, while milk feeding rate equals the average daily milk allowance/birth body weight (%). ^4^ Outcomes = effect of forage inclusion, ** indicates a significant effect (*p* < 0.05), * indicates a tendency (*p* < 0.1), N indicates a negative effect, *p* indicates a positive effect, and NS shows no significant effect. DMI and ADG were evaluated by comparing calves fed with or without forage.

**Table 4 animals-10-00188-t004:** A summary of studies differing in methods and time of offering forage in pre-weaned dairy calves.

Objectives	Trt ^1^	Calf/Trt	Weaning Age (d)	Forage Feeding Age (d)	Forage Source	Forage (%)	Cutting Length/Processing ^2^	Concentrate Physical Form	Amount of Milk Fed ^3^	Outcomes ^4^		Reference
Forage Feeding Method/Time										DMI	ADG	
Sole vs. TMR Free choice-	4	12	50	1	Grass hay	TMR (0, 15%), Free choice	Chopped <2.5 cm	Texture	534 L (Around 26%)	**NS**	**NS**	*Overvest* et al., 2015 [77]
Sole vs. TMR, Free choice	3	15	57	3	Alfalfa hay	TMR (0, 10%), Free choice	GMPL: 3 mm	Finely ground	262 L (Around 11%)	***p* ****	**NS**	*EbnAli* et al., 2016 [78]
Sole vs. Free choice	3	60	56	4	Alfalfa hay, Oats hay	Free choice	Chopped to approximately 2.5 cm	Pellet	376 L (Around 17%)	**NS**	**NS**	*Xiao* et al., 2018 [25]
Day 14, 28, 42	4	10	57	-	Alfalfa hay	TMR (0, 15%)	GMPL: 3 mm	Ground	Around 10%	***p* ****	***p* ****	*Hosseini* et al., 2015 [75]
Day 3, 15	5	8	56	-	Alfalfa hay, Oats hay	Free choice	Chopped	Pellet	358 L (Around 16%)	**NS**	**NS**	*Wu* et al., 2018 [24]
Day 14, 42	3	6	63	-	Oat hay	Free choice	-	-	252 L (Around 10%)	***p* ****	***p* ****	*Lin* et al., 2018 [60,76]

^1^ Trt = Treatment. ^2^ GMPL refers to geometric mean particle length, which was determined using ASABE forage sieve methods (ANSI/ASAE S424.1) [45]. ^3^ Amount of milk fed is the total amount fed during the whole trial period, while milk feeding rate equals the average daily milk allowance/birth body weight (%). ^4^ Outcomes = effect of forage inclusion, ** indicates a significant effect (*p* < 0.05), N indicates a negative effect, *p* indicates positive effect, and NS shows no significant effect. DMI and ADG were evaluated by comparing calves fed with or without forage.

**Table 5 animals-10-00188-t005:** A summary of studies on forage inclusion in dairy calves based on different physical forms of concentrate.

Objectives	Trt ^1^	Calf/Trt	Weaning Age (d)	Forage Feeding Age (d)	Forage Source	Forage (%)	Forage Cutting Length/Processing ^2^	Solid Feed Offering Method	Amount of Milk Fed ^3^	Outcomes ^4^		Reference
Physical Form of Concentrate										DMI	ADG	
Pellet, Mash	4	16	28	3	Beet pulp	0, 16%	-	TMR	80 L (Around 7%)	**NS**	**NS**	*Porter* et al., 2007 [37]
Pellet, Texture	3	11	49	7	Ryegrass hay	Ad libitum (0, 6.8, 11.6%)	-	Free choice	274 L (Around 16%)	**NS**	**NS**	*Terré* et al., 2015 *(Trail 1)* [80]
Pellet, Texture	3	20	52	8	Ryegrass hay	Ad libitum (0, 4.3%)	-	Free choice	233 L (Around 13%)	**NS**	**NS**	*Terré* et al., 2015 *(Trail 2)* [80]
Mash, Texture	4	12	56	3	Corn silage	0, 15%	GMPL: 0.5, 1.1, 3.0, and 4.0 mm	TMR	291 L (Around 13%)	***p* ****	***p* ****	*Mirzaei* et al., 2016 [67]

^1^ Trt = Treatment. ^2^ GMPL refers to geometric mean particle length, which was determined using ASABE forage sieve methods (ANSI/ASAE S424.1) [45]. ^3^ Amount of milk fed is the total amount fed during the whole trial period, while milk feeding rate equals the average daily milk allowance/birth body weight (%). ^4^ Outcomes = effect of forage inclusion, ** indicates a significant effect (*p* < 0.05), N indicates a negative effect, *p* indicates positive effect, and NS shows no significant effect. DMI and ADG were evaluated by comparing calves fed with or without forage.

**Table 6 animals-10-00188-t006:** A summary of studies on forage inclusion in dairy calves based on different volumes and methods of milk feeding.

Objectives	Trt ^2^	Calf/Trt	Weaning Age (d)	Forage Feeding Age (d)	Forage Source	Forage (%)	Cutting Length/Processing ^3^	Solid Feed Offering Method	Concentrate Physical Form	Outcomes ^4^		Reference
Milk Feeding Amount/Method ^1^										DMI	ADG	
359 L (Around 16%), 221 L (Around 10%)	4	8	56	1	Beet pulp	Beet pulp (0%, 18%)	Chopped	Free choice	Pellet	**NS**	**N ***	*Kosiorowska* et al., 2010 [84]
350 L (Around 20%)	2	15	56	3	Orchard grass hay	Ad libitum	Chopped	-	-	**NS**	**NS**	*Khan* et al., 2011 [20]
534 L (Around 26%)	4	12	50	1	Grass hay	TMR (0, 15%), Free choice-ad libitum	Chopped < 2.5 cm	Free choice; TMR	Texture	**NS**	**NS**	*Overvest* et al., 2015 [77]
212 L (Around 9%), 338 L (Around 15%)	6	10	56	4	Wheat straw	0, 7.5, 15%	-	TMR	Ground	***p* ***	***p* ***	*Hosseini* et al., 2019 [40]
Step Down vs. Conventional; 313 L (Around 13%,)	4	20	60	3	Alfalfa hay	0, 15%	-	TMR	Finely ground	***p* ****	***p* ****	*Daneshvar* et al., 2015 [23]
Teat vs. Bucket; 241 L (Around 13%)	3	10	45 ± 2	1–3	Timothy hay	Ad libitum	Chopped around 5 cm	-	Pellet	**NS**	**NS**	*Horvath* et al., 2017 [85]

^1^ The amount of milk fed is the total amount fed during the whole trial period, while the milk feeding rate equals the average daily milk allowance/birth body weight (%). ^2^ Trt = Treatment. ^3^ GMPL refers to geometric mean particle length, which was determined using ASABE forage sieve methods (ANSI/ASAE S424.1) [45]. ^4^ Outcomes = effect of forage inclusion, ** indicates a significant effect (*p* < 0.05), * indicates a tendency (*p* < 0.1), N indicates a negative effect, *p* indicates positive effect, and NS shows no significant effect. DMI and ADG were evaluated by comparing calves fed with or without forage.

**Table 7 animals-10-00188-t007:** Summary of selected studies on forage inclusion and their effect on performance, rumen fermentation, and expressive behavior compared to calves fed only concentrates ^1^.

Parameters ^2^	Studies with Positive Impact	Studies with Negative Impact	Studies with No Effect
Total solid DMI	[15,18,19,20,21,22,23,38,39,40,43,50,51,55,67,75,76,78,80]	[5,15,17,39,43]	[21,24,25,26,37,44,48,49,50,52,54,56,57,77,80,84,86]
ADG	[18,19,20,21,22,23,38,39,40,50,51,53,55,67,75,76,80]	[5,15,17,39,84]	[15,21,24,25,26,37,43,44,48,49,50,52,56,57,77,78,80,85]
DM digestibility	/	[23,37,41,48]	[21,39,52,75,78]
Feed to Gain ratio	[19,39]	[5,15,22,51]	[21,23,24,26,27,37,38,39,40,44,49,52,53,55,56,67,75,78,80,85]
Structural growth	[55]	[15]	[20,23,25,27,38,39,40,44,49,57,67,76,80]
Fecal score	/	[27,37]	[19,26,39,44,49,75]
Rumen fluid pH	[20,22,23,25,27,37,38,39,40,41,42,52,53,54,55,67,75,76,78,80]	/	[5,24,39,43]
Total VFA	[22]	[19,23,25,35,40,52,53,54,67]	[24,27,38,39,42,43,76,78]
Acetate	[22,23,40,43,52,53,67]	/	[19,24,27,38,39,76,78]
Propionate	[53]	[25,35,40,43,54,67]	[19,22,23,24,27,38,39,52,76,78]
Butyrate	/	[23,25,27,39,40,53]	[19,22,24,38,39,43,52,54,67,76,78]
Valerate	/	[23,25,52,53,67]	[40,76]
Acetate to Propionate ratio	[27,37,38,40,42,54,67]	[53]	[19,22,23,24,39,52]
Lactate	/	[51]	[43]
NH_3_	[42]	[43]	[24]
Rumen papillae length	/	[42,44,52]	[20,22,27,37,43,76,84]
Rumen plaque formation	/	[22,43]	[42]
Rumen weight	[11,20,50]	[27,43]	[24,52,76,84]
Rumen volume	[11,42,52]	/	[24,76]
Ruminating	[21,39,40,53,55,56,75,76,77,78]	/	[39]
Total eating behavior	[39,40,57,76,77]	/	[39,55,78,85]
Concentrate eating behavior	[57]	[56,85]	[21,53,75]
Drinking behavior	[76]	[56]	/
Non-nutritive oral behavior/Abnormal behavior	/	[21,40,53,55,56,57,76,85]	[75,78]
Lying behavior	/	[21,53,75,76]	[39,40,55,56,77,78]
Standing behavior	[76]	/	[21,39,40,53,55,56,75,78]
Satisfaction behavior	[56]	/	[57]
Urination and Defecation behavior	/	/	[56]
Sorting behavior	/	[25,87]	[77]

^1^ Forages included dry hay, silage, straw, and by-products (e.g., cottonseed hulls) and reconstituted hay. Positive, negative, and no effect on a parameter was determined by adding forage in the diet compared with no forage inclusion in those studies. “Positive effect” represent an increase or improved effect (*p* < 0.05), “Negative effect” represent a decreased effect in the related parameter (*p* < 0.05), “/” means no studies were found to affect this parameter in the current review (*p* > 0.05).^2^ Parameters were measured in dairy calves within 3 months of age. Non-nutritive oral behavior/abnormal behavior included tongue rolling, licking buckets, pen or surface, sniffing, vocalizing, and eating beddings; Satisfaction behavior included tail swishing, self-grooming, and rubbing.

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
