# Peer review of "Review: How Forage Feeding Early in Life Influences the Growth Rate, Ruminal Environment, and the Establishment of Feeding Behavior in Pre-Weaned Calves"

_animals, 2020, doi:10.3390/ani10020188_

Round 1

Reviewer 1 Report

The manuscript needs considerable amount of editing of the English

More care needs to be taken to edit down and reference each statement of a study.

Table 1 should have been split into appropriate subject areas according to the main objectives and then each table should be reorganized within each table to demonstrate the main outcomes of the studies.

Author Response

Responses to the Manuscript animals-668691: Review: How forage feeding early in life influences the growth performancerate, ruminal environment and the behavioral establishment of feeding behavior by in dairy pre-weaned calves

Dear Editor and Reviewers,

We take this opportunity to thank you for the time you have taken to review our manuscript and the invaluable comments you have given us. We have, to the best of our knowledge, made the necessary changes to improve the quality of this manuscript from its current form. Please find below our response to your comments.

Reviewer 1:

The manuscript needs considerable amount of editing of the English

AU: We have thoroughly revised the the manuscript to make it more readable and improve the quality.

More care needs to be taken to edit down and reference each statement of a study.

AU: We have re-referenced the statements as suggested.

Table 1 should have been split into appropriate subject areas according to the main objectives and then each table should be reorganized within each table to demonstrate the main outcomes of the studies.

AU: We split Table 1 into several tables that capture the key findings of either offering or not offering forage on calf performance.

Reviewer 2 Report

This review was well designed and provides important information and a comprehensive overview of forage inclusion in young calves. The review was generally well-written with few grammatical errors and was well organized. I recommend accept this review after revise some minor suggestions that list below:

Line 25: delete “try and”

Line 55: delete “.”

Line 75: change “behavior” to “behavior establishment”

Line 93: change “rate” to “percentage”

Line 95: change “calves’” to “calves”

Line 96: “hence the proliferation of related research is conducted”

Line 98: change “a study” to “studies”

Line 103: change to “the hay intake decreased linearly, and the live and empty BW and rumen development increased”

Line 121: delete “with”

Line 131: need to add the “percentage of alfalfa hay has been used”

Line 135: delete “(EBWADG)”

Line 179: change “wheat straw” to “wheat straw:”

Line 183: delete “added”

Line 230: delete “contrary to previous studies.”

Line 252: change to “affect behavior development and expression in dairy calves”

Line 269: change “a” to “another”

Line 298: add “lower” before fermentation rate

Line 313: this sentence needs to be revised

Line 359: change “,” to ”,”

Line 415: change to “accumulation was inconsistent [104] or transiently fluctuated [105] in lactic acid”

Line 459-461: delete or rewrite this sentence

Author Response

Responses to the Manuscript animals-668691: Review: How forage feeding early in life influences the growth performancerate, ruminal environment and the behavioral establishment of feeding behavior by in dairy pre-weaned calves

Dear Editor and Reviewers,

We take this opportunity to thank you for the time you have taken to review our manuscript and the invaluable comments you have given us. We have, to the best of our knowledge, made the necessary changes to improve the quality of this manuscript from its current form. Please find below our response to your comments.

Reviewer 2:

Line 25: delete “try and”

AU: Changed

Line 55: delete “.”

AU: Changed

Line 75: change “behavior” to “behavior establishment”

AU: Changed

Line 93: change “rate” to “percentage”

AU: Changed

Line 95: change “calves’” to “calves”

AU: Changed

Line 96: “hence the proliferation of related research is conducted”

AU: Changed

Line 98: change “a study” to “studies”

AU: Changed

Line 103: change to “the hay intake decreased linearly, and the live and empty BW and rumen development increased”

AU: Changed

Line 121: delete “with”

AU: Changed

Line 131: need to add the “percentage of alfalfa hay has been used”

AU: Changed

Line 135: delete “(EBWADG)”

AU: Changed

Line 179: change “wheat straw” to “wheat straw:”

AU: Changed

Line 183: delete “added”

AU: Changed

Line 230: delete “contrary to previous studies.”

AU: Changed

Line 252: change to “affect behavior development and expression in dairy calves”

AU: Changed

Line 269: change “a” to “another”

AU: Changed

Line 298: add “lower” before fermentation rate

AU: Changed

Line 313: this sentence needs to be revised

AU: Changed

Line 359: change “,” to ”,”

AU: Changed

Line 415: change to “accumulation was inconsistent [104] or transiently fluctuated [105] in lactic acid”

AU: Changed

Line 459-461: delete or rewrite this sentence

AU: Changed

Reviewer 3 Report

There was a lot of work in this review. Good job.

Author Response

Responses to the Manuscript animals-668691: Review: How forage feeding early in life influences the growth rate, ruminal environment and the behavioral establishment of feeding behavior by in dairy pre-weaned calves

Dear Editor and Reviewers,

We take this opportunity to thank you for the time you have taken to review our manuscript and the invaluable comments you have given us. We have, to the best of our knowledge, made the necessary changes to improve the quality of this manuscript from its current form. Please find below our response to your comments.

Reviewer 3:

The authors are commended on their work in reviewing this vast area of literature. Although there was a recent review done by Imani et al. (2017) and Diao et al. (2019), this review adds different factors that may have an interaction with forage, such as sorting behavior.

Although most of the manuscript was well written, there are significant areas that should be revised. Next time, I would highly suggest having a native speaker read the manuscript before submission.

Throughout the manuscript, I would suggest changing “pre-weaning calves” to pre-weaned calves. This is a preference and not necessary for acceptance.

AU: Changed “pre-weaning calves” to pre-weaned calves throughout the manuscript.

Line 16: add comma after contrary

AU: Changed

Line 17: add “the” in front of ration

AU: Changed

Line 19: change “the young calves’ diets” to young calf diets or diets of young calves.

AU: Changed

Line 28: change calf’s to calf

AU: Changed

Line 28: add comma after forage and alfalfa (line 29)

AU: Changed

Line 47: change “as it was believed that they could reduce abnormal growth and behavior in fed milk alone” to believed to reduce abnormal growth and behavior, lower…

AU: Changed

Line 48: change “plays a vital role in rumen development” to improve rumen development

AU: Changed

Line 54-55: Incomplete sentence. Add “It was found that high concentrate diets produce higher levels of butyrate and propionate”

AU: This sentence has been modified.

Line 60: change provide to providing

AU: This sentence has been modified.

Line 62: change calf to calves

AU: Corrected

Line 66: Change the calves’ performance to calf performance

AU: This sentence has been modified.

Line 66: take out “the” before lower concentrate consumption

AU: This sentence has been modified.

Line 68-71: Rewrite with better English grammar

AU: This sentence has been modified.

Line 76: Change to: Factors that affect calf performance with forage inclusion

AU: Changed

Line 83: add “only allowed” before free access since the original sentence makes it sound like commercial dairy farms feed calves without forage and free access to starter feed

AU: This sentence has been modified.

Line 85: Change inclusion to including

AU: This sentence has been modified.

Line 85: change calves’ diet to calf diet or diet of calves

AU: Changed

Line 87: change calf’s to calf

AU: This sentence has been modified.

Line 91: In Table 1, is the Objective column necessary with so little data? I would take that out since it doesn’t provide much information.

AU: Table 1 has been modified, it has been split into several subject areas according to the main objectives and then each table has been reorganized to show the main outcomes of the studies.

Table 1 reports all studies included in this review. However, some recent studies are missing such as Guzman et al. (2015), Kehoe et al. (2019), Dill-McFarland et al. (2017), and Xia et al. (2018). Some of these have input on the section 3.2

AU: Table 1 did not include all studies, it summarized the studies which investigate the effect of forage inclusion on pre-weaned calf performance, it mainly separated and discussed the factors affecting forage use in dairy calves. The study of Xia et al (2018) was not included, because weaned calves (156.8 kg) was used in his study. The study of Guzman et al. (2015) was not included because all calves are euthanized before d 3 of age and no growth performance data was measured.

The research of Dill-McFarland et al. (2019) and Kehoe et al. (2019) has been added in Table 1 and 3.

Line 95: Change calves’ to calf diets

AU: Changed

Line 97: change calf ration to calf rations

AU: Changed to diets

Line 104: change “excellent” to a different word

AU: Changed “more excellent” to “improved”

Line 105: change concentrate to concentrations

AU: Changed

Line 105: change enhances to enhance

AU: Changed

Line 109: take out “the” before total nutrient

AU: Changed

Line 110: change “of forage” to “forage inclusion

AU: Changed

Line 111: change diet on calves to diet on calf performance

AU: Changed

Line 113: change differences to difference

AU: Changed

Line 116: In these studies, ADG and BW were higher but bulk effect should be mentioned. More hay intake, more bulkiness (and potentially rumen mass and muscle) which would increase ADG and BW more than calves not given forage.

AU: Bulk effect has been mentioned at the end of this paragraph as you suggested.

Line 117: change “to” to “in”

AU: This sentence has been modified.

Line 133: Change “on including forage in the diet” to “from forage inclusion”

AU: This sentence has been modified.

Line 134: change “have shown” to showed

AU: Changed to reported

Line 136: Hill used low-quality chopped timothy hay which is the reason given for poor performance. However, in the paragraph earlier, Hosseini et al. used wheat straw which also should be low quality for preweaned dairy calves. Is there another reason these studies found different results?

AU: Both Castells and Hosseini supplemented straw in the diets and found a positive effect on calf performance, the inconsistent result on calf performance of providing low quality hay was discussed in the end of this paragraph, it might relate to the different amount of milk offered in Hill, Castells and Hosseini study.

Line 138: change calves’ to calf

AU: Changed

Line149-151: Explain this statement more: alfalfa not exhibiting the same benefits due to free access of the forage. This sentence leaves the reader wondering why would free access and palatable hay not provide the same benefits as the other forages.

AU: This sentence has been modified.

Line 177: change “in” to “by”

AU: Changed

Line 178: remove “studies.”

AU: Changed

Line 183: use either mixed or added.

AU: Changed to mixed

Line 190: change moisturizing to moisturized

AU: Changed

Line 190: “reconstituting the solid feed” should be explained in a different way. Maybe “rehydrating hay”?

AU: This sentence has been modified.

Line 203-204: this has been done in at least one recently published study (Kehoe et al, 2019; Dill-McFarland, 2017)

AU: The research of Kehoe et al, 2019 have been added. This sentence has been modified.

Line 217: add comma after corn silage, reconstituted hay

AU: Changed

Line 218: add “s” to calf diet

AU: Changed

Line 229-232: Not supported fully since just talked about the Wu study which found the opposite. Also, feed costs may actually increase if DMI increases due to forage inclusion.

AU: This sentence has been rewritten.

Line 238-243: Rewrite, very confusing

AU: This paragraph has been rewritten.

Line 243-253: Rewrite, poor grammar.

AU: This sentence has been rewritten.

Line 256: Remove spacing after first line

AU: Removed

Line 271: Replace “the calves’” with calf

AU: Changed

Line 274: this increases saliva production to neutralize rumen pH and acids

AU: This sentence has been added.

Line 290: add “the” to “in Hill et al. study

AU: Changed

Line 292: remove “wholly”

AU: Changed

Line 296: Hill et al. is not citation #86

AU: The citation has been added.

Line 298: “have fermentation rates” is not a complete thought.

AU: This sentence has been modified. Lower has been added in front of the fermentation rate.

Line 298: change “the low” to low. This allows the reader to assume there are not only low energy forages but also other (higher energy) forages. Don’t you love English grammar?

AU: Changed, thanks for your kindly remind

Line 303: change “alter” to a different word to make more sense

AU: This sentence has been modified.

Line 304-307: The latter group was the high milk allowance which says has the higher forage selection rate? The next few sentences go on to say that lower milk consumption will increase concentrate and forage intake. Please rewrite.

AU: This paragraph has been rewritten.

Line 316:  change” the former had better performance” to “found that the former…”

AU: This sentence has been modified.

Line 329: take out “the” before total DMI…and before forage

AU: Changed

Line 330: change “to the one with” to “from”

AU: Changed

Line 340-342: Rewrite, poor grammar.

AU: This sentence has been rewritten.

Line 345: The rumen

AU: Changed

Line 358: What is the 84% and 16%?

AU: Explanation has been added.

Line 359: What was found lacking?

AU: This sentence has been modified.

Line 361: change “in calf” to “to calves”

AU: Changed

Line 365: change provide to provided

AU: Changed

Line 368: change “supplementing chopped oat hay which contains more effective fiber could…” to “supplementing chopped oat hay, containing more effective fiber, could…”

AU: This sentence has been modified.

Line 369-372: Run-on sentence. Rewrite

AU: This sentence has been modified.

Line 382: Besides is not a great word to be using often. Change to a different word (maybe “Also”).

AU: Changed

Line 386: the rumen wall instead of rumen wall

AU: Changed

Line 387: from the rumen lumen instead of from rumen lumen

AU: Changed

Line 392: in the rumen microbial ecosystem.

AU: Changed

Line 400: anything past 10 years probably should not be considered recent. Remove 37 and 38 citation.

AU: This sentence has been deleted.

Line 401: What does “supplementing forage inconsistently” mean? Please explain more or differently.

AU: This sentence has been deleted.

Line 412: remove “when”

AU: Removed

Line 413: change “that led to” to “leading to”

AU: Changed

Line 413: Change comma to period

AU: Changed

Line 419-20: change affects to affected and alter to altered.

AU: Changed

Line 427-434: there are a lot of unnecessary “the”

AU: Changed

Line 436: change “young calves microbiota” to “the microbiota of young calves.”

AU: Changed

Line 453: fragment sentence

AU: This sentence has been deleted.

Line 454 and Line 462: If Prevotella is a dominant genus of Bacteroidetes, then these two lines say the exact opposite (Bacteroidetes decrease with concentrate feeding and Prevotella increase with concentrate feeding). Please clarify.

AU: This part has been modified.

It was demonstrated that the relative abundance of Bacteroidetes dropped in adult animals fed concentrate feed and they were repeatedly induced with SARA [112]. The lower level of Bacteroidetes in calves fed only concentrate could be partly explained by a higher concentrate feed intake and a lower pH in the rumen. Furthermore, Kim et al. [44] found that the most dominant genus of Bacteroidetes was Prevotella, which is highly active hemicellulolytic and starch-degraders [114] that mainly produce acetate. A relatively higher abundance of Prevotella may be related to the higher acetate proportion in forage offering groups compared to no forage groups [44]. Similarly, Jami et al [106] have reported that Prevotella was observed to be the predominant genus in animals fed high-fiber diets rather than high-calorie diets.

Line 472: change “in a different solid consumption” to “in different solid feed consumption”

AU: This sentence has been modified.

Line 474: add “the” in front of Lin et al. study

AU: Changed

Line 478: change comma to period

AU: Changed

Line 484: change to “to understand better substrate fermentation”

AU: Changed to “to better understand substrate fermentation”

Line 485: change “when” to “as”

AU: Changed

Line 490: change “recent 10 years” to “latest decade”

AU: Changed

Line 491: change calves’ to calf

AU: Changed

Line 495: change spend to spent

AU: Changed

Line 497: take out “the” before newborn ruminants

AU: Removed

Line 500: change feeding to receiving

AU: Changed

Line 500: take out “Besides”

AU: Removed

Line 501: change “physical” to “physically” and “but” to “and”

AU: Changed

Line 503: take out “the” in front of saliva production and VFA

AU: Changed

Line 518: change induce to inducing

AU: Changed

Line 521: take out “the” in front of post-ingestive feedback

AU: Changed

Line 523: Change to “Feed sorting is also seen in the early life of calves, when feeding…”

AU: This sentence has been modified.

Line 536: change suggested to suggest

AU: Changed

Line 541: months to month

Line 550: change claves to calves

AU: Changed

Line 555: change “buckets, pen or surface licking” to “licking of…”

AU: Changed

Line 559: change “besides”

AU: Changed to furthermore

Line 556: Is allogrooming considered a negative behavior? Maybe “excessive allogrooming”?

AU: Deleted. Based on the previous study (phillips et al. 2004), allogrooming can in some circumstances be considered as a problematic behavior. However, it is a relative word and we can’t quantify to what extent it is problematic. We have thus deleted the word.  

Line 573: What is “expression” referring to?

AU: This sentence has been modified.

Line 580: Change “literatures” to “research articles”

AU: Changed

Line 676: Nemati is 2016 but referred to 2017

AU: Changed

Line 692-3: formatting

AU: Changed

Line 745: Different year cited

AU: Changed